# QUANTUM ALGORITHM FOR SPARSE ONLINE LEARNING WITH TRUNCATED GRADIENT DESCENT

## ABSTRACT

Logistic regression, the Support Vector Machine (SVM) and least squares are well-studied methods in the statistical and computer science community, with various practical applications. High-dimensional data arriving on a real-time basis makes the design of online learning algorithms that produce sparse solutions essential. The seminal work of Langford et al. (2009) developed a method to obtain sparsity via truncated gradient descent, showing a near-optimal online regret bound. Based on this method, we develop a quantum sparse online learning algorithm for logistic regression, the SVM and least squares. Given efficient quantum access to the inputs, we show that a quadratic speedup in the time complexity with respect to the dimension of the problem is achievable, while maintaining a regret of $O(1/\sqrt{T})$, where $T$ is the number of iterations.

## 1 INTRODUCTION

The field of statistical modeling and machine learning is about developing robust methodologies to uncover patterns, make predictions, and derive insights from data. Three prominent techniques are logistic regression, the support vector machines (SVMs) and least squares. Regression and data fitting make use of least squares (Hansen et al., 2013; Eberly, 2000; Cantrell, 2008; Watson, 1967; Geladi and Kowalski, 1986; Audibert and Catoni, 2011; Maillard and Munos, 2009; Boyd and Vandenberghe, 2018) to study the relationship between predictor variables and a response variable in a set of data. In particular, given $N$ data points and their labels $\{x^{(i)}, y^{(i)}\}_{i=1}^N$ such that $x^{(i)} \in \mathbb{R}^d$ and $y^{(i)} \in \mathbb{R}$ for all $i \in [N]$[1], and a model function $f : \mathbb{R}^d \to \mathbb{R}$, the goal is to find the optimal $w$ that minimizes the squared loss $\sum_{i=1}^N (f(x^{(i)}, w) - y^{(i)})^2$.

Unlike linear regression which models continuous outcomes, logistic regression is adept at predicting the probability of a discrete outcome, for example success or failure, making it an essential tool for understanding and predicting categorical data (LaValley, 2008; Nick and Campbell, 2007; Menard, 2002; Das, 2021; Sperandei, 2014; Stoltzfus, 2011). In short, logistic regression can be described as follows: given a set of $N$ data points and their labels $\{x^{(i)}, y^{(i)}\}_{i=1}^N$ such that $x^{(i)} \in \mathbb{R}^d$ and $y^{(i)} \in \{-1, 1\}$ for all $i \in [N]$, logistic regression aims to find a $w \in \mathbb{R}^d$ that minimizes the loss $\sum_{i=1}^N \ln(1 + e^{-(w \cdot x^{(i)}) \cdot y^{(i)}})$.

On the other hand, SVM is a widely used tool in machine learning and finds applications in the domain of chemistry, biology, finance (Ivanciuc, 2007; Huang et al., 2018; Yang, 2004; Tay and Cao, 2001), due to its simplicity of use and robust performance. A support vector machine is an algorithm that classifies vectors in a feature space into one of two sets, given training data from the sets (Cortes and Vapnik, 1995). SVM works by constructing the optimal hyperplane that partitions the two sets, either in the original feature space or a higher-dimensional kernel space. Given a set of $N$ data points and their labels $\{x^{(i)}, y^{(i)}\}_{i=1}^N$ such that $x^{(i)} \in \mathbb{R}^d$ and $y^{(i)} \in \{-1, 1\}$ for all $i \in [N]$, the goal is to find a $w \in \mathbb{R}^d$ that minimizes the hinge loss $\sum_{i=1}^N \max\{0, (1 - y^{(i)} w \cdot x^{(i)})\}$.

**The online learning framework**    Online learning algorithms have gained much attention in recent decades, in both the academic and industrial sectors (Hoffman et al., 2010; Bottou, 1998; Kivinen et al., 2004; Dekel et al., 2012; Helmbold et al., 1998; Anava et al., 2013; Hoi et al., 2021). In

---

[1] $[N]$ denotes the set $\{1, \cdots, N\}$.

this framework, the learner (also known as the learning algorithm) who is given access to partial, sequential training data, is required to output a solution based on partial knowledge of the training data. The solution is then updated in the next iteration after receiving more training data as input. This process is repeated for $T$ number of iterations. More specifically, for every time $t = 1, \cdots, T$, the following sequence of events take place: 1) The learner receives an unlabelled example $x^{(t)}$; 2) The learner makes a prediction $\hat{y}^{(t)}$ based on an existing weight vector $w^{(t)} \in \mathbb{R}^d$; 3) The learner receives the true label $y^{(t)}$ and suffers a loss $L(w^{(t)}, x^{(t)}, y^{(t)})$ that is convex in $w^{(t)}$; 4) The learner updates the weight vector according to some update rule $w^{(t+1)} \leftarrow f(w^{(t)})$.

Due to the fact that the input data of online algorithms can be adversarial in nature, such algorithms are particularly useful in proving guarantees for worst-case inputs. Besides having an efficient running time, the design of online algorithms focuses on regret minimization. The *regret* of an online algorithm is defined as the difference between the total loss incurred using a certain sequence of strategies and the total loss incurred using the best fixed strategy in hindsight (Hazan et al., 2007). Specifically,[2]

$$Regret := \frac{1}{T} \sum_{t=1}^{T} L\left(w^{(t)}, x^{(t)}, y^{(t)}\right) - \min_{u \in \mathbb{R}^d} \frac{1}{T} \sum_{t=1}^{T} L\left(u, x^{(t)}, y^{(t)}\right).$$

In the era of big data, we are often faced with large and high-dimensional problem data. As a result, the solution to the learning problem will inherit a large dimension. Techniques such as best subsets, forward selection, and backward elimination (Mao, 2002; 2004; Whitley et al., 2000; Ververidis and Kotropoulos, 2005; Borboudakis and Tsamardinos, 2019; Reif and Shafait, 2014; Tan et al., 2008; Zongker and Jain, 1996; Kumar and Minz, 2014; Wei and Billings, 2006; Dai and Wen, 2018; Peng and Linetsky, 2022; Zhang et al., 2015) enhance computational efficiency and ease the interpretability of the solution. Seeing the importance of sparse solutions and the strength of online algorithms, the need for sparse online learning is apparent (Liang and Poon, 2021; Wang et al., 2015; Lin et al., 2016).

Langford et al. (2009) introduced a truncated gradient descent algorithm for sparse online learning. In their algorithm, the solution is updated via gradient descent at every iteration and a truncation is performed on the solution after every $K$ iterations, where $K$ needs to be carefully chosen (see Algorithm 4 in Appendix A.1). In their work, the following assumptions are made.

**Assumption 1** (Langford et al. (2009)). *For every $t \in [T]$,*

(i) *The loss function $L(w^{(t)}, x^{(t)}, y^{(t)})$ is convex in $w^{(t)}$ for all $x^{(t)}, y^{(t)}$.*

(ii) *There exist constants $A, B \in \mathbb{R}_{\geq 0}$ such that $\|\nabla_{w^{(t)}} L(w^{(t)}, x^{(t)}, y^{(t)})\|_2^2 \leq A \cdot L(w^{(t)}, x^{(t)}, y^{(t)}) + B$ for all $x^{(t)}, y^{(t)}$, where $\| \cdot \|_2$ denotes the Euclidean norm.*

(iii) *$\sup_{x^{(t)}} \|x^{(t)}\|_2 \leq C$ for some constant $C \in \mathbb{R}_+$.*

Under these assumptions, the authors of Langford et al. (2009) showed that their online algorithm achieves an $O(1/\sqrt{T})$ regret (refer to Appendix A.2). As also noted in Langford et al. (2009), the general loss function for linear prediction problems is of the form $L(w^{(t)}, x^{(t)}, y^{(t)}) = h(w^{(t)\mathsf{T}} x^{(t)}, y^{(t)})$. They pointed out some common loss functions $h(\cdot, \cdot)$ with corresponding choices of parameters $A$ and $B$ (which are not necessarily unique), under the assumption that $\sup_{x^{(t)}} \|x^{(t)}\|_2 \leq C$. Among them are

- Logistic regression: $h(w^{(t)\mathsf{T}} x^{(t)}, y^{(t)}) = \ln(1 + \exp(-w^{(t)\mathsf{T}} x^{(t)} \cdot y(t)))$; $A = 0$, $B = C^2$, $y^{(t)} \in \{\pm 1\}$ for all $t \in [T]$.

- SVM (hinge loss): $h(w^{(t)\mathsf{T}} x^{(t)}, y^{(t)}) = \max\{0, 1 - w^{(t)\mathsf{T}} x^{(t)} \cdot y(t)\}$; $A = 0$, $B = C^2$, $y^{(t)} \in \{\pm 1\}$ for all $t \in [T]$.

- Least squares (square loss): $h(w^{(t)\mathsf{T}} x^{(t)}, y^{(t)}) = (w^{(t)\mathsf{T}} x^{(t)} - y^{(t)})^2$; $A = 4C^2$, $B = 0$, $y^{(t)} \in \mathbb{R}$ for all $t \in [T]$.

---

[2]Strictly speaking, this is the per-step regret as we normalize by $T$. While the conventional regret is the unnormalized version, we nevertheless call this the regret in this paper.

The motivation of our work is three-fold: First, high-dimensional data calls for the need for sparse solutions. Second, high-dimensional data often present the challenge of limited sample sizes, a scenario common in fields such as bioinformatics, finance, and image processing. This situation, commonly referred to as the "high-dimension, low sample size" (HDLSS) problem, creates significant difficulties for traditional statistical and machine learning methods (Fan and Li, 2006; Hastie, 2009). More specifically, one may have access to only a small sample of high-dimensional data. This could be due to the scarcity of data sources such as in the study of rare diseases, costly data acquisition, storage limitations, experiments being carried out in controlled settings where increasing sample size might not be feasible due to strict experimental conditions (Consortium et al., 2015; Mitani and Haneuse, 2020; Konietschke et al., 2021). In such cases, an efficient algorithm that effectively learns sparse solutions is crucial. Third, the evolution of quantum computing has gained much traction in the recent years, bringing about provable speedups. From the celebrated Grover's algorithm for unstructured search to Shor's factoring algorithm, there is a wide range of quantum algorithms that improve over their classical counterparts (Nielsen and Chuang, 2010; Grover, 1996; Shor, 1999). Seeing wide applications of logistic regression, SVM and least squares, we are motivated to devise a quantum algorithm that learns sparse solution in the online setting.

**Main contribution** Our main contribution is a quantum online algorithm that outputs a sparse solution, which has applications to logistic regression (Section 5.1), the SVM (Section 5.2) and least squares (Section 5.3). We realize how to *systematically* trade-off the update and the prediction, whereas previous approaches considered each problem individually. Moreover, we have obtained further speedups in certain tasks over previous approaches. Our work is based on Langford et al. (2009), who introduced a truncated gradient descent algorithm for sparse online learning. The guarantees of our algorithm hold under the same assumption 1 as Langford et al. (2009) for all cases, with an additional assumption 2 that applies specifically to least squares. While maintaining the $O(1/\sqrt{T})$ regret bound of Langford et al. (2009), our quantum algorithm has time complexity of $\tilde{O}(T^{5/2}\sqrt{d})^3$, achieving a quadratic speedup in the dimension over the classical $O(Td)$, where $d$ is the dimension of a data point. This speedup is noticeable when $d \geq \tilde{\Omega}(T^5)$, making the algorithm useful for high-dimensional learning tasks. We summarize our results in the Table 1.

Table 1: Summary of results

| Problem | Time Complexity | | Regret | |
|---|---|---|---|---|
| | Langford et al. (2009) | **Our Work** | Langford et al. (2009) | **Our Work** |
| Logistic regression | $O(Td)$ | $\tilde{O}(T^{5/2}\sqrt{d})$ | $O(1/\sqrt{T})$ | $O(1/\sqrt{T})$ |
| SVM | $O(Td)$ | $\tilde{O}(T^{5/2}\sqrt{d})$ | $O(1/\sqrt{T})$ | $O(1/\sqrt{T})$ |
| Least squares | $O(Td)$ | $\tilde{O}(T^{5/2}\sqrt{d})$ | $O(1/\sqrt{T})$ | $O(1/\sqrt{T})$ |

Our algorithm does not need to read in all the entries of the input data at once. For a data point $x^{(t)} \in \mathbb{R}^d$, the $j$-th entry of $x^{(t)}$, $x_j^{(t)}$, can be accessed in $\tilde{O}(1)$ time on a classical computer. We assume that $x_j^{(t)}$ can be accessed in $\tilde{O}(1)$ time coherently on a quantum computer (formally defined in Data Input 1), which is the *standard quantum input model* employed in the previous literature, e.g., Grover (1996); Li et al. (2019); Brandão et al. (2019).

Our quantum algorithm returns the weight vectors $w^{(1)}, w^{(2)}, \ldots, w^{(T)}$ indirectly. Specifically, for each $w^{(t)}$ with $1 \leq t \leq T$, our algorithm enables us to *coherently* access each of its entries in $O(t)$ time. Notably, the cost of accessing one entry of $w^{(t)}$ is upper bounded by $O(T)$, and $T$ is usually set to $O(1/\epsilon^2)$ (which is *independent* of the dimension $d$) if we want a regret of $\epsilon$ in concrete applications such as logistic regression, SVM, and least squares in Table 1. Especially for constant regret, e.g., $\epsilon = 0.1$, the cost is a constant time. In addition, this output model is even useful when we do not need to know the exact value of each entry of $w^{(t)}$ but certain expectations with respect to $w^{(t)}$. As suggested in Harrow et al. (2009), the (normalized) quantum state $|w^{(t)}\rangle$ can be useful in estimating the expectations. To this end, our quantum algorithm allows us to further prepare $|w^{(t)}\rangle$

---
[3] $\tilde{O}(\cdot)$ suppresses polylogarithmic factors.

with an extra time complexity of $\tilde{O}(t\sqrt{d})$ through the standard quantum state preparation (Grover, 2000); this is negligible compared to the overall time complexity $\tilde{O}(T^{5/2}\sqrt{d})$.

**Techniques** Our quantum algorithm is based on the framework of Langford et al. (2009). In this framework, the algorithm maintains a sparse weight vector by performing the truncation regularly. Our main observation is that in the framework, a large number of updates (linear in the data dimension $d$) are required (on a classical computer) while the prediction in each iteration is just a single real number. This motivates us to find a reasonable trade-off between the update and the prediction, and we then realized how to achieve this on a quantum computer.

Our quantum speedup comes from the techniques that rely on quantum amplitude estimation and amplification (Brassard et al., 2002; Harrow and Wei, 2020; Rall and Fuller, 2023; Cornelissen and Hamoudi, 2023). In particular, we use subroutines such as quantum inner product estimation, quantum norm estimation and quantum state preparation. This allows us to obtain a quadratic speedup in the dimension $d$ for the prediction. For the update, we do not actually perform the updates but implement them in an oracle-oriented manner so that any entry of the intermediate vectors can be computed in $\tilde{O}(T)$ time, which is sufficient for us to make the prediction efficiently on a quantum computer. Specifically, we leverage the circuits for efficient arithmetic operations to avoid storing the weight vector in every iteration, thereby saving the space and time of the algorithm. Under this quantized framework, we develop quantum algorithms for logistic regression, the SVM and least squares by specifying the quantum circuits with appropriate parameters for the corresponding problems.

**Applications** Let $u^*$ be the best fixed strategy in hindsight. For logistic regression and the SVM, we observe that by taking $T = \Theta(C^4\|u^*\|_2^4/\epsilon^2)$, the regret of our quantum algorithm becomes $\Theta(\epsilon)$. This implies quantum algorithms for (offline) logistic regression and SVM with time complexity $\tilde{O}(C^{10}\|u^*\|_2^{10}\sqrt{d}/\epsilon^5)$. Our algorithm is obtained under a unified framework, though our time complexity is slightly worse than the prior best one for logistic regression due to Shao (2019). Nevertheless, our algorithm achieves a polynomial improvement in the $\epsilon$-dependence for the SVM, compared to the prior best offline result due to Li et al. (2019). Taking $T$ to be the aforementioned value for the SVM also results in a $\Theta(\epsilon)$ regret and a time complexity of $\tilde{O}(C^6\|u^*\|_2^6\sqrt{d}/\epsilon^3)$. This implies a quantum algorithm for (offline) SVM that achieves a polynomial improvement in the dependence on $\epsilon$ as compared to the existing best offline result Li et al. (2019). Moreover, taking $T = \Theta((C^6 + C^4\|u^*\|_2^4)/\epsilon^2)$ results in a $\Theta(\epsilon)$ regret for least squares and a time complexity of $\tilde{O}(C^{15}\|u^*\|_2^6\sqrt{d}/\epsilon^5)$, if the prediction error is constant-bounded. This implies a quantum algorithm for (offline) least squares. For reference, a quantum algorithm for offline least squares with different conditions was presented in Liu and Zhang (2017).[4]

## 2 RELATED WORK

Motivated by the importance of obtaining sparse solutions for the aforementioned reasons in the previous section, various methods have been used to achieve sparsity in learning algorithms. Some examples include randomized rounding (Golovin et al., 2013), forward sequential selection (Cotter et al., 2005), backward sequential elimination (Cotter et al., 2001) and adaptive forward-backward greedy algorithm (Zhang, 2008). Some variants of truncation methods have also been used such as high order truncated gradient descent being applied to ridge regression in Li et al. (2020a), median-truncated gradient descent where only samples whose absolute values are not too deviated from the sample median are included (Chi et al., 2019; Li et al., 2020b; Khonglah and Mukherjee, 2023), truncated regression, truncated kernel stochastic gradient descent (Bai and Shi, 2024), truncated regression (Daskalakis et al., 2019; 2020; 2021),

In the online setting, a method called Forward-Backward Splitting (FOBOS) with an $\ell_1$-norm regularizer has been proposed by Duchi and Singer (2009). This approach is analogous to the projected gradient descent, where the projection step is replaced with a minimization problem. Inspired by

---

[4]In these offline settings, the output is the "average" vector $\bar{w} = \frac{1}{T}\sum_{t=1}^{T} w^{(t)}$. For a fair comparison with the offline results, after the execution of our algorithm, it provides quantum access to each entry of $\bar{w}$ at an extra cost of $\tilde{O}(T^2)$, where $T$ is usually chosen to be independent of the dimension $d$ in the presented applications.

dual-averaging techniques, the Regularized Dual Averaging (RDA) algorithm by Xiao (2009) is designed for problems whose objective function consists of a convex loss function and a convex regularization term. Under certain conditions, it achieves $O(1/\sqrt{T})$ regret and proves effective for sparse online learning with $\ell_1$-regularization. Wang et al. (2013) gave two online feature selection algorithms, which are modifications of the Perceptron algorithm (Rosenblatt, 1958). The performance of both algorithms is evaluated numerically and via the mistake bound. Other references on sparse online learning include Zhao et al. (2020a); Mairal et al. (2010); Ma and Zheng (2017); Song et al. (2019); Hao et al. (2021); Liu et al. (2019b).

In the offline setting, quantum algorithms for least squares and SVM have been studied. The recent work by Song et al. (2023) gave a quantum algorithm that outputs a solution such that the $\ell_2$-norm of the residual vector approximates that of the optimal solution up to a relative error of $\epsilon > 0$ with high probability. Their algorithm runs in time $\tilde{O}(\sqrt{n}d^{1.5}/\epsilon + d^\omega/\epsilon)$, where $n$ is the sample complexity and $\omega \approx 2.37$ denotes the exponent of matrix multiplication. This improves upon the best classical algorithm which runs in time $O(nd) + \text{poly}(d/\epsilon)$ (Clarkson and Woodruff, 2017). Moreover, Liu and Zhang (2017) proposed a quantum algorithm that solves the same problem in time $O\left(\log(n+d)s^2\kappa^3/\epsilon^2\right)$, where $s$ denotes the sparsity of the data matrix and $\kappa$ is the condition number. Other quantum algorithms for linear regression such as Refs. Wang (2017); Kerenidis and Prakash (2017); Chakraborty et al. (2019) demonstrate that exponential quantum speedups are achievable. However, the time complexity of these algorithms depends on some quantum linear-algebra related parameters, such as the condition number of the data matrix. For the SVM, Rebentrost et al. (2014) proved exponential advantage compared to any known classical algorithm for certain data sets. For the case of general data, Li et al. (2019) gave a quantum algorithm that runs in time $\tilde{O}\left(\frac{\sqrt{n}}{\epsilon^4} + \frac{\sqrt{d}}{\epsilon^8}\right)$, improving over the classical running time of $O\left(\frac{n+d}{\epsilon^2}\right)$ by Clarkson et al. (2012). The complexity of the quantum SVM has been studied by Gentinetta et al. (2024). Several sublinear time quantum algorithms were developed under the framework of online learning, e.g., for semidefinite programming (Brandão et al., 2019; van Apeldoorn et al., 2020; Brandao and Svore, 2017), zero-sum games (Li et al., 2019; van Apeldoorn and Gilyén, 2019; Bouland et al., 2023; Gao et al., 2024), general matrix games (Li et al., 2021) and learning of quantum states (Aaronson et al., 2018; Yang et al., 2020; Chen et al., 2024).

With regards to how quantum computing can improve the efficiency of algorithms on feature selection, Saeedi and Arodz (2019) proposed the Quantum Sparse Support Vector Machine (QsSVM), an approach that minimizes the training-set objective function of the Sparse SVM model (Bennett, 1999; Kecman and Hadzic, 2000; Bi et al., 2003; Zhu et al., 2003) by using a quantum algorithm for solving linear programs (LPs) (van Apeldoorn and Gilyén, 2019) instead of a classical LP solvers. While quantum LP solvers may not speed up arbitrary binary classifiers, they offer sublinear time complexity in the number of samples and features for sparse linear models, unlike classical algorithms. Sampling can be considered a form of feature selection. The quantum online portfolio optimization algorithm by Lim and Rebentrost (2024) employs quantum multi-sampling to invest in sampled assets. Their approach achieves quadratic speedup compared to classical methods (Helmbold et al., 1998), with a marginal increase in regret. Lin et al. (2020) studied quantum-enhanced least-square SVM with two quantum algorithms. The first employs a simplified quantum approach using continuous variables for matrix inversion, while the second is a hybrid quantum-classical method providing sparse solutions with quantum-enhanced feature maps, both achieving exponential speedup in sample size. Other related quantum algorithms include Liu et al. (2019a); Wang and Xiang (2019); Rebentrost et al. (2014); Doriguello et al. (2023); Liu and Zhang (2017). Recent advancement in quantum algorithms can be found in survey papers (Bacon and VAn DAm, 2010; Biamonte et al., 2017; Zhang and Ni, 2020; Mishra et al., 2021; Dalzell et al., 2023).

## 3 PRELIMINARIES

**Quantum computing** In classical computing, the basic unit of information is a bit, which can take values 0 or 1. In quantum computing, the basic unit is known as a quantum bit, or *qubit*. It is a two-level quantum system with states $|0\rangle = (1 \quad 0)^T$ and $|1\rangle = (0 \quad 1)^T$. Unlike a classical bit that has only two states, a qubit is a *superpositions* of $|0\rangle$ and $|1\rangle$, i.e. $|v\rangle = \sum_{i=0}^{1} v_i |i\rangle$, where $v_i \in \mathbb{C}$ is the *amplitude* of $|i\rangle$ and satisfies $\sum_{i=0}^{d-1} |v_i|^2 = 1$. The states $|0\rangle, |1\rangle$ forms

the (orthogonal) computational basis of the two-dimensional Hilbert space. This extends to any $d$-dimensional system, where $d > 2$. Quantum states from different Hilbert spaces can be combined using tensor product. For simplicity of notation, we use $|u\rangle |v\rangle$ to denote the tensor product $|u\rangle \otimes |v\rangle$. Operations in quantum computing are *unitary*, i.e. a linear transformation $U$ that satisfies $UU^\dagger = U^\dagger U = I$, where $U^\dagger$ is the conjugate transpose of $U$.

The information in a quantum state cannot be "read" directly. In order to observe a quantum state $|v\rangle$, we perform a *quantum measurement* on it. The measurement results in a classical state $i$ with probability $|v_i|^2$, and the measured quantum state *collapses* to $|i\rangle$. Quantum access to input data is encoded in a unitary operator known as the *quantum oracle*. Quantum oracles allow data to be accessed in superposition, thereby allowing operations to be performed "simultaneously" on states, which is the core of quantum speedups.

**Notations** For a positive integer $d \in \mathbb{Z}_+$, we use $[d]$ to represent the set $\{1, \cdots, d\}$. Given a vector $u \in \mathbb{R}^d$, we denote the $j$-th entry of $u$ as $u_j$ for all $j \in [d]$ and denote the $\ell_1$-norm and $\ell_\infty$ norm of $u$ as $\|u\|_1 := \sum_{j=1}^d |u_j|$ and $\|u\|_\infty := \max_{j \in [d]} |u_j|$. If the vector has a time dependency, we denote it as $u^{(t)}$. For some condition $C$, we use $I(C)$ to denote the indicator function that evaluates to 1 if $C$ is satisfied and 0 otherwise. We use $\bar{0}$ to denote the all zeros vector and use $|\bar{0}\rangle$ to denote the state $|0\rangle \otimes \cdots \otimes |0\rangle$ where the number of qubits is clear from the context. We use $\tilde{O}(\cdot)$ to hide polylog factors, i.e. $\tilde{O}(f(n,m)) = O(f(n,m) \cdot \text{polylog}(n,m))$.

**Quantum computational model** A quantum algorithm is described by a quantum circuit with queries to the input oracle. We define the query complexity of a quantum algorithm as the number of queries to the input oracle. The time complexity of a quantum algorithm is the sum of its query complexity and the number of elementary quantum gates in it. We assume a quantum arithmetic model, which allows us to ignore issues arising from the fixed-point representation of real numbers. In this model, each elementary arithmetic operation takes a constant time. Our quantum algorithm assumes quantum query access to the input oracle for certain vectors. For a vector $u \in \mathbb{R}^d$, the input oracle for $u$ is a unitary operator $O_u$ such that $O_u : |j\rangle |\bar{0}\rangle \to |j\rangle |u_j\rangle$ for every $j \in [d]$, where the second register is assumed to contain sufficiently many qubits to ensure that all subsequent computations are accurate, in analogy to the sufficient bits that a classical algorithm assumes to run correctly.

**Truncated gradient descent** One of the most popular approaches for minimizing a convex loss function $L$ is gradient descent. Starting from an initial point $w^{(1)}$, gradient descent performs $w^{(t+1)} = w^{(t)} + \eta^{(t)} \nabla L(w^{(t)})$ for $t = 1, 2, 3, \cdots$, where $\eta^{(t)} > 0$ is the step size/learning rate at iteration $t$ and $\nabla L(w^{(t)})$ is the gradient of $L$ at $w^{(t)}$. The algorithm is summarized in Algorithm 2 (refer to Appendix A.1). Seeing the need for sparse solutions in the high-dimensional regime, Langford et al. (2009) introduced a truncated gradient descent update rule that truncates each entry $j \in [d]$ of the weight vector after certain number of iterations according to the following function: for some threshold $\theta > 0$ and a *gravity* parameter[5] $\alpha > 0$,

$$\mathcal{T}(w_j^{(t)}, \alpha, \theta) = \begin{cases} \max\{w_j^{(t)} - \alpha, 0\}, \text{ if } 0 \le w_j^{(t)} \le \theta \\ \min\{w_j^{(t)} + \alpha, 0\}, \text{ if } -\theta \le w_j^{(t)} \le 0 \\ w_j^{(t)}, \text{ otherwise.} \end{cases} \tag{1}$$

The truncated gradient descent method is summarized in Algorithm 3 (refer to Appendix A.1).

While adaptive learning rates could potentially speed up the convergence of gradient descent (Grimmer, 2023; Zeiler, 2012; Malitsky and Mishchenko, 2023), we adopt a constant learning rate with fixed $\eta > 0$ as in the setting of Langford et al. (2009) for simplicity. Moreover, while the choice of gravity parameters is usually kept open in practice, we shall only consider the following choice: for all $t \in [T]$, $\alpha^{(t)} = g^{(t)} \eta$ such that $g^{(1)} = \cdots = g^{(T)} \le g_{\max}$, where $g_{\max}$ is some constant.

---

[5]The gravity parameter measures the amount of shrinkage.

## 4 QUANTUM ALGORITHM

In this section, we present our quantum algorithm for sparse online learning, which has applications to logistic regression, the SVM and least squares. We clarify the data input and output model, as well as quantum subroutines in the next subsections.

### 4.1 QUANTUM INPUT AND OUTPUT MODEL

We assume quantum query access via an oracle for the entries of unlabelled examples. The online nature of the problem is given by the fact that we obtain these oracles at different times.

**Data Input 1** (Online example oracles). *Let $x^{(1)}, \cdots, x^{(T)} \in \mathbb{R}^d$ be unlabelled examples. Define the unitary $U_{x^{(t)}}$ operating on $O(\log d)$ qubits such that for all $j \in [d]$ and $t \in [T]$, $U_{x^{(t)}} |j\rangle |\bar{0}\rangle \to |j\rangle |x_j^{(t)}\rangle$. At time $t \in [T]$, assume access to $U_{x^{(1)}}, \cdots, U_{x^{(t)}}$.*

As for the output of our quantum algorithm, it returns weight vectors $w^{(1)}, w^{(2)}, \ldots, w^{(T)}$ indirectly. In particular, for all $t \in [T]$, we are allowed coherent access to the entries of $w^{(t)}$ in $O(t)$ time. This cost is upper bounded by $O(T)$. In addition, this output model allows us to further prepare $|w^{(t)}\rangle$, through the standard quantum state preparation (Grover, 2000), as suggested in Harrow et al. (2009), such a (normalized) quantum state $|w^{(t)}\rangle$ can be useful in estimating expectations.

### 4.2 QUANTUM SUBROUTINES

By approximating real numbers to sufficient accuracy, the truncation function Eq. (1) can be efficiently computed by assuming access to *minmax* and *between* oracles. Similar oracles have been studied in Refs. Vedral et al. (1996); Ambainis (2007); Addanki et al. (2021); Luongo et al. (2024).

**Definition 1** (Minmax and Between oracles (Luongo et al., 2024)). *Let $a, b, x \in \{0, 1\}^n$ and $c, z \in \{0, 1\}$.*

*(i) We say that we have access to a comparison oracle $\mathcal{O}_{comp}$ if we have access to a unitary $U_{comp}$ that performs the operation $U_{comp} : |x\rangle |a\rangle |z\rangle \mapsto |x\rangle |a\rangle |z \oplus \mathbb{1}_{x<a}\rangle$ using $s_{comp}$ Toffoli gates.*

*(ii) We say that we have access to a controlled-comparison oracle $\mathcal{O}'_{comp}$ if we have access to a unitary $U'_{comp}$ that performs the operation $U'_{comp} : |c\rangle |x\rangle |a\rangle |z\rangle \mapsto |c\rangle |x\rangle |a\rangle |z \oplus c \cdot \mathbb{1}_{x<a}\rangle$ using $s'_{comp}$ Toffoli gates.*

*(iii) Assuming access to a comparison oracle and a controlled-comparison oracle, there exists a circuit that performs the operation $U_{Btw} : |a\rangle |b\rangle |x\rangle |z\rangle \mapsto |a\rangle |b\rangle |x\rangle |z \oplus \mathbb{1}_{x \in [a,b]}\rangle$ using $1.5 s_{comp} + s'_{comp}$ Toffoli gates. We cal this the between oracle.*

*The values of $s_{comp}$ and $s'_{comp}$ depend on the type of circuit architecture used for the comparators (Gidney, 2018; Cuccaro et al., 2004). Nevertheless, these are in general $O(n)$. We say that we have access to a minmax oracle if we have access to a unitary $U_{minmax}$ that performs the following operation*

$$U_{Minmax} |c\rangle |x\rangle |0\rangle = \begin{cases} |c\rangle |x\rangle |\max(x,0)\rangle, & \text{if } c = 1 \\ |c\rangle |x\rangle |\min(x,0)\rangle, & \text{if } c = 0. \end{cases}$$

*using $O(n)$ number of Toffoli gates.*

Using the oracles defined above, we show the following lemma, whose proof can be found in Appendix A.3.

**Lemma 1** (Truncation unitary). *Let $\theta, \alpha \in \mathbb{R}_{>0}$. Assuming access to a Between oracle and a Minmax oracle, there exists a unitary operator $U_{\mathcal{T}, \alpha, \theta}$ that does the following operation up to sufficient accuracy in constant time: $U : |x\rangle |0\rangle \mapsto |x\rangle |f(x)\rangle$, where*

$$f(x) = \begin{cases} \max\{x - \alpha, 0\}, & 0 < x \le \theta, \\ \min\{x + \alpha, 0\}, & -\theta \le x < 0, \\ x, & \text{otherwise.} \end{cases}$$

Unbiased amplitude estimation (Harrow and Wei, 2020; Rall and Fuller, 2023; Cornelissen and Hamoudi, 2023) allows one to obtain nearly unbiased estimates with low variance and without destroying their initial quantum state.

**Fact 1** (Unbiased amplitude estimation; Theorem 2.4, (Cornelissen and Hamoudi, 2023))**.** *Let* $t \geq 4$ *and* $\epsilon \in (0, 1)$. *We are given one copy of a quantum state* $|\psi\rangle$ *as input, as well as a unitary transformation* $U = 2 |\psi\rangle \langle\psi| - I$, *and a unitary transformation* $V = I - 2P$ *for some projector* $P$. *There exists a quantum algorithm that outputs* $\tilde{a}$, *an estimate of* $a = \|P |\psi\rangle\|^2$, *such that* $|\mathbb{E}[\tilde{a}] - a| \leq \epsilon$ *and* $\mathrm{Var}[\tilde{a}] \leq \frac{91a}{t^2} + \epsilon$ *using* $\mathcal{O}(t \log \log(t) \log(t/\epsilon))$ *applications of* $U$ *and* $V$ *each. The algorithm restores the quantum state* $|\psi\rangle$ *at the end of the computation with probability at least* $1 - \epsilon$.

Quantum state preparation, norm and inner production are widely used subroutines in References: van Apeldoorn and Gilyén (2019); Brassard et al. (2002); Hamoudi et al. (2019); Li et al. (2019); Rebentrost et al. (2021) that rely on amplitude estimation (Brassard et al., 2002). We restate these subroutines for the convenience of the reader. The proof is based on Fact 1, which is deferred to Appendix A.4. We also note that an additive variant of Lemma 2(i) can be easily derived with the same time complexity and resources.

**Lemma 2** (Quantum norm estimation and state preparation (van Apeldoorn and Gilyén, 2019; Brassard et al., 2002; Hamoudi et al., 2019; Li et al., 2019; Rebentrost et al., 2021))**.** *Let* $u \in \mathbb{R}^d$ *and assume quantum access to* $u \in \mathbb{R}^d$. *Then,*

  (i) *Let* $\delta \in (0, 1/4)$ *and* $\epsilon_0 \in (0, 1)$. *There exists a quantum algorithm that outputs an estimate* $\tilde{\Gamma}$ *of* $\|u\|_1$ *such that* $|\tilde{\Gamma} - \|u\|_1| \leq \epsilon_0 \|u\|_1$ *with probability at least* $1 - 4\delta$. *The algorithm runs in time* $\tilde{O}\left(\frac{\sqrt{d}}{\delta}\right)$.

  (ii) *Let* $\zeta \in (0, 1/2]$ *and* $\tilde{\Gamma} > 0$ *be given such that* $|\|u\|_1 - \tilde{\Gamma}| \leq \zeta \|u\|_1$. *Let* $\delta \in (0, 1)$. *An approximation* $|\tilde{p}\rangle = \sum_{j=1}^d \sqrt{|p_j|} |j\rangle$ *of the state* $|u\rangle = \sum_{j=1}^d \sqrt{\frac{|u_j|}{\|u\|_1}} |j\rangle$ *can be prepared with probability* $1 - \delta$ *in* $O\left(\sqrt{d} \log \frac{1}{\delta}\right)$ *time and* $\tilde{O}\left(\sqrt{d} \log \frac{1}{\zeta} \log \frac{1}{\delta}\right)$ *gates. The approximation in* $\ell_1$*-norm of the probabilities is* $\|\tilde{p} - \frac{u}{\|u\|_1}\|_1 \leq 2\zeta$

The following lemma explains a quantum inner product estimation algorithm which uses amplitude estimation to approximate the inner product between two real vectors. We defer the proof to Appendix A.5.

**Lemma 3** (Quantum inner product estimation (Yang et al., 2023))**.** *Let* $\epsilon, \delta \in (0, 1)$ *and given access to nonzero vectors* $u, v \in \mathbb{R}^d$. *An estimate* $\widetilde{IP}$ *of the inner product* $u \cdot v$ *can be obtained such that* $|\widetilde{IP} - u \cdot v| \leq \epsilon$ *with success probability at least* $1 - \delta$. *This requires* $O\left(\frac{\|u\|_\infty \|v\|_1 \sqrt{d}}{\epsilon} \log \frac{1}{\delta}\right)$ *queries and* $\tilde{O}\left(\frac{\|u\|_\infty \|v\|_1 \sqrt{d}}{\epsilon} \log \frac{1}{\delta}\right)$ *quantum gates.*

### 4.3 QUANTUM ALGORITHM FOR SPARSE ONLINE LEARNING

Given Data Input 1, the following computation can be performed in superposition on indices $j \in [d]$, which allows us to efficiently compute each entry of the weight vector $w^{(t)}$ at any time step $t$. We defer the proof to Appendix A.6. Similar unitaries were studied in, e.g., References Chakrabarti et al. (2021); Li et al. (2019); Rebentrost et al. (2021); Vedral et al. (1996).

**Lemma 4.** *Let* $\theta \in \mathbb{R}_{>0}$. *For all* $t \in [T]$, *given example oracles* $U_{x^{(t)}}$ *as in Data Input 1, vectors* $y = (y^{(1)}, \cdots, y^{(t)}), \tilde{y} = (\tilde{y}^{(1)}, \cdots, \tilde{y}^{(t)}) \in \mathbb{R}^t$ *and a real number* $\eta \in \mathbb{R}_{>0}$. *Assuming access to a gravity sequence* $(g^{(1)}, \cdots, g^{(T)})$ *and a truncation oracle as in Lemma 1, there exists a unitary operators that perform the operation* $|j\rangle |\bar{0}\rangle \to |j\rangle |w_j^{(t)}\rangle$ *to sufficient numerical precision, where*

  (i) *For logistic regression,* $w_j^{(t)} = \mathcal{T}\left(w_j^{(t-1)} + 2\eta \frac{x_j^{(t)} y^{(t)} e^{-y^{(t)} \tilde{y}^{(t)}}}{1 + e^{-y^{(t)} \tilde{y}^{(t)}}}, g^{(t)} \eta, \theta\right);$

(ii) *For the SVM,* $w_j^{(t)} = \begin{cases} \mathcal{T}\left(w_j^{(t)} + \eta y^{(t)} x_j^{(t)}, g^{(t)}\eta, \theta\right), & \text{if } y^{(t)}\tilde{y}^{(t)} < 1 \\ \mathcal{T}\left(w_j^{(t)}, g^{(t)}\eta, \theta\right), & \text{otherwise} \end{cases}$ ;

(iii) *For least squares,* $w_j^{(t)} = \mathcal{T}\left(w_j^{(t)} + 2\eta\left(y^{(t)} - \tilde{y}^{(t)}\right)x_j^{(t)}, g^{(t)}\eta, \theta\right).$

*This computation takes $O(T)$ queries to the data input and requires $O(T + \log d)$ qubits and quantum gates.*

We present our quantum algorithm for sparse online learning in Algorithm 1. Note that this quantum algorithm applies to logistic regression, the SVM and least squares when choosing their respective unitaries from Lemma 4 in Line 5.

---

**Algorithm 1** Quantum algorithm for sparse online learning with truncated gradient descent

---

   **Input:** Threshold $\theta > 0$, gravity sequence $\{g^{(1)}, \cdots, g^{(T)}\} \leq g_{\max}$, learning rate $\eta \in (0, 1)$, $\tilde{y}^{(1)} = 0$, failure probability $\delta$, errors $\epsilon_{\mathrm{IP}}, \epsilon_{\mathrm{norm}} \in (0, 1)$.
   **Output:** $|\tilde{w}^{(1)}\rangle, \cdots, |\tilde{w}^{(T)}\rangle$.
1: **for** $t = 1$ to $T$ **do**
2:     Receive example oracle $U_{x^{(t)}}$.
3:     Compute the estimate $\tilde{y}^{(t)}$ of the inner product $\hat{y}^{(t)} = \sum_{j=1}^{d} w_j^{(t)} x_j^{(t)}$ up to additive accuracy
         $\epsilon_{\mathrm{IP}}$ with success probability $1 - \frac{\delta}{3T}$ using Lemma 3.
4:     Receive the true label $y^{(t)}$.
5:     Prepare the state $|w^{(t+1)}\rangle$ with success probability $1 - \frac{\delta}{3T}$ using Lemma 1, 2(ii) and Lemma 4
         (depending on the problem).
6:     Obtain an estimate $\tilde{q}^{(t+1)}$ of $\|w^{(t)} \cdot I\left(\left|w^{(t)}\right| \leq \theta\right)\|_1$ using Lemma 2(i) up to additive error
         $\epsilon_{\mathrm{norm}}$ with success probability $1 - \frac{\delta}{3T}$.
7: **end for**

---

We emphasize that each entry of the weight vector can be efficiently computed, independently of the other entries as shown in Lemma 4. This includes the truncation step. Hence, it follows that the weight vectors need not be stored in qubits, thereby saving on the space/memory of the algorithm. For the inner product estimation, we access the entries of the weight vector by simply computing them, thanks to efficient unitaries for arithmetic computation and truncation. These entries, after the truncation, will not be entangled with auxiliary qubits.

## 5 CONVERGENCE AND TIME COMPLEXITY ANALYSIS

In this subsection, we analyze the regret and time complexity of Algorithm 1 applied to logistic regression, SVM, and least squares, respectively. We show that our quantum algorithms achieve a quadratic speedup in the data dimension $d$ over classical algorithms.

### 5.1 QUANTUM SPARSE ONLINE ALGORITHM FOR LOGISTIC REGRESSION

The quantum sparse online algorithm for logistic regression is obtained by applying Algorithm 1 with Lemmas 4(i). The regret is guaranteed by Theorem 1, with its proof deferred to Appendix A.7.

**Theorem 1** (Regret for online logistic regression). *Let $\delta \in (0, 1)$. For all $t \in [T]$, let $\tilde{y}^{(t)}$ be an estimate of $\hat{y}^{(t)} = w^{(t)\mathsf{T}} x^{(t)}$ to additive error $\epsilon_{\mathrm{IP}}$, and $\tilde{q}^{(t+1)}$ be an estimate of $q^{(t+1)} := \|w^{(t+1)} \cdot I(|w^{(t+1)}| \leq \theta)\|_1$ to additive error $\epsilon_{\mathrm{norm}}$. Set $\eta = \frac{1}{C^2\sqrt{T}}$, $\epsilon_{\mathrm{norm}} = \frac{1}{2\eta T}$ and $\epsilon_{\mathrm{IP}} = \frac{1}{2\sqrt{T}}$. Under Assumption 1(iii), Algorithm 1 with Lemma 4(i) achieves a regret bound of*

$$\frac{1}{T}\sum_{t=1}^{T}\ln\left(1 + e^{-y^{(t)}\tilde{y}^{(t)}}\right) + \frac{1}{T}\sum_{t=1}^{T}g^{(t)}\tilde{q}^{(t)} - \frac{1}{T}\sum_{t=1}^{T}\ln\left(1 + e^{-y^{(t)}u^{\mathsf{T}}x^{(t)}}\right)$$

$$-\frac{1}{T}\sum_{t=1}^{T}g^{(t)}\left\|u \cdot I\left(\left|w^{(t+1)}\right|\right)\right\|_1 \leq \frac{1 + C^2\left(2 + g_{\max} + \|u^*\|_2^2\right)}{2\sqrt{T}}$$

*with success probability $1 - \delta$ using $O\left(T^{5/2}\sqrt{d}\log\frac{T}{\delta}\right)$ queries, where $g_{\max} = \max_{t\in[T]} g^{(t)}$.*

*Proof Sketch:* Denote the regret from Langford et al. (2009) be denoted as $R_c$ and the regret of our quantum algorithm as $R_q$. We use Lipchitz-continuity, Assumption 1(iii), Lemma 2(i) and 3 to bound the the difference $R_q - R_c$. Lastly, the total regret of the quantum algorithm is bounded using triangle inequality $R_q \leq |R_q - R_c| + |R_c|$. □

By taking $T = \Theta(C^4\|u^*\|_2^4/\epsilon^2)$, the regret in Theorem 1 becomes $\Theta(\epsilon)$. This implies a quantum algorithm for (offline) logistic regression with time complexity $\tilde{O}(C^{10}\|u^*\|_2^{10}\sqrt{d}/\epsilon^5)$. However, the performance of our algorithm is not better than the prior best offline result is $\tilde{O}(C^3\|u^*\|_2^2/\epsilon^2)$ due to Shao (2019).

## 5.2 QUANTUM SPARSE ONLINE ALGORITHM FOR SVM

The quantum sparse online algorithm for SVM is obtained by applying Algorithm 1 with Lemma 4(ii). The regret is guaranteed by Theorem 2, with its proof deferred to Appendix A.8.

**Theorem 2** (Regret for online hinge loss). *Let $\delta \in (0,1)$. For all $t \in [T]$, let $\tilde{y}^{(t)}$ be an estimate of $\hat{y}^{(t)} = w^{(t)\mathsf{T}}x^{(t)}$ to additive error $\epsilon_{\text{IP}}$ and $\tilde{q}^{(t+1)}$ be an estimate of $q^{(t+1)} := \left\|w^{(t+1)} \cdot I\left(\left|w^{(t+1)}\right| \leq \theta\right)\right\|_1$ to additive error $\epsilon_{\text{norm}}$. Set $\eta = \frac{1}{C^2T^2}$ and $\epsilon_{\text{IP}} = \epsilon_{\text{norm}} = \frac{1}{2\sqrt{T}}$. Under Assumption 1(iii), Algorithm 1 with Lemma 4(ii) achieves a regret bound of*

$$\frac{1}{T}\sum_{t=1}^{T}\left(1 - y^{(t)}\tilde{y}^{(t)}\right)^+ + \frac{1}{T}\sum_{t=1}^{T}g^{(t)}\tilde{q}^{(t+1)} - \frac{1}{T}\sum_{t=1}^{T}\left(1 - y^{(t)}u^\mathsf{T}x^{(t)}\right)^+$$

$$-\frac{1}{T}\sum_{t=1}^{T}g^{(t)}\left\|u \cdot I\left(\left|w^{(t+1)}\right|\right)\right\|_1 \leq \frac{2 + C^2\left(g_{\max} + \|u^*\|_2^2\right)}{2\sqrt{T}}$$

*with success probability $1 - \delta$ using $O\left(T^{5/2}\sqrt{d}\log\frac{T}{\delta}\right)$ queries, where $g_{\max} = \max_{t\in[T]} g^{(t)}$.*

*Proof Sketch:* Same as Theorem 1 . □

By taking $T = \Theta(C^4\|u^*\|_2^4/\epsilon^2)$, the regret in Theorem 2 becomes $\Theta(\epsilon)$. This implies a quantum algorithm for (offline) SVM with time complexity $\tilde{O}(C^{10}\|u^*\|_2^{10}\sqrt{d}/\epsilon^5)$, achieving a better dependence on $\epsilon$ than the prior best offline result $\tilde{O}(\sqrt{d}C^2\|u^*\|_2^2/\epsilon^5 + \sqrt{d}/\epsilon^8)$ due to Li et al. (2019).

## 5.3 QUANTUM SPARSE ONLINE ALGORITHM FOR LEAST SQUARES

The quantum sparse online algorithm for least squares is obtained by applying Algorithm 1 with Lemma 4(iii). It turns out that the quantum speedup appears when the prediction error is constant-bounded. Here, the prediction error means the distance between true and predicted labels, which was also considered previously in the literature, e.g., Lin et al. (2022). We formally state this condition as follows.

**Assumption 2** (Constant-bounded prediction error). *For any $t \in [T]$, let $y^{(t)}$ be the true label and let $\hat{y}^{(t)} = \sum_{j=1}^{d} w_j^{(t)} \cdot x_j^{(t)}$ be the predicted label. The prediction error is bounded by a constant $D \in \mathbb{R}_+$, such that for all $t$, $\left|y^{(t)} - \hat{y}^{(t)}\right| \leq D$.*

Under Assumption 2, the regret is guaranteed by Theorem 3, with its proof in Appendix A.9.

**Theorem 3** (Regret for online least squares). *Let $\delta \in (0,1)$. Let $u \in \mathbb{R}^d$ be any vector and for $t \in [T]$, let $\tilde{y}^{(t)}$ be an estimate of $\hat{y}^{(t)} = w^{(t)\mathsf{T}}x^{(t)}$ to additive error $\epsilon_{\text{IP}}$ and $\tilde{q}^{(t+1)}$ be an estimate of $q^{(t+1)} := \left\|w^{(t+1)} \cdot I\left(\left|w^{(t+1)}\right| \leq \theta\right)\right\|_1$ to additive error $\epsilon_{\text{norm}}$. Set $\eta = \frac{1}{C^2\sqrt{T}}$, $\epsilon_{\text{IP}} = \frac{1}{2\sqrt{T}}$ and $\epsilon_{\text{norm}} = \frac{1}{2\eta T}$. Under Assumptions 1(iii) and 2, Algorithm 1 with Lemma 4(iii) achieves a regret bound of*

$$\frac{1}{T}\sum_{t=1}^{T}\left(\tilde{y}^{(t)} - y^{(t)}\right)^2 + \frac{1}{T}\sum_{t=1}^{T}g^{(t)}\tilde{q}^{(t+1)} - \frac{1}{T}\sum_{t=1}^{T}\left(u^\mathsf{T}x^{(t)} - y^{(t)}\right)^2$$

$$-\frac{1}{T}\sum_{t=1}^{T} g^{(t)} \left\| u \cdot I\left(\left|w^{(t+1)}\right| \leq \theta\right)\right\|_1 \leq \frac{C^2\left(CD + g_{\max} + \|u^*\|_2^2\right)}{\sqrt{T}}$$

*with probability at least* $1 - \delta$ *using* $O\left(T^{5/2}\sqrt{d}\log\frac{T}{\delta}\right)$ *queries, where* $g_{\max} = \max_{t\in[T]} g^{(t)}$.

*Proof Sketch:* Same as Theorem 1, with the additional application of Assumption 2. $\qquad\square$

By setting $T = \Theta((C^6 + C^4\|u^*\|_2^4)/\epsilon^2)$, the regret in Theorem 3 becomes $\Theta(\epsilon)$. This implies a quantum algorithm for (offline) least squares with time complexity $\tilde{O}((C^{15} + C^{10}\|u^*\|_2^{10})\sqrt{d}/\epsilon^5)$. For comparison, we are aware of a quantum algorithm for offline least squares proposed in Liu and Zhang (2017) that considers different conditions and parameters. Their algorithm has time complexity $\tilde{O}(s^2\kappa^2/\epsilon^2)$, where $s$ denotes the sparsity of the data matrix and $\kappa$ is its condition number.

## 6    DISCUSSION AND CONCLUSION

We propose a quantum online learning algorithm that outputs sparse solutions. Our quantum algorithm can be applied to logistic regression, the SVM and least squares. We show that the quantum algorithm achieves a quadratic speedup in the dimension of the problem as compared to its classical counterpart. The speedup stems from the use of quantum subroutines based on quantum amplitude estimation and amplification. We note that the speedup is only noticeable when $d \geq \Omega(T^5\log^2(T/\delta))$, which makes the algorithm useful for high-dimensional learning tasks. As our quantum algorithm is erroneous, it is natural that convergence is achieved after a greater number of steps as compared to its classical analogue. Our algorithm maintains a regret bound of the same order as compared to the classical algorithm of Langford et al. (2009), i.e. $O(1/\sqrt{T})$. We leverage unitaries that perform arithmetic computations, which allows us to save on the space/memory of the algorithm for storing the weight vector, which is $O(d)$ in Langford et al. (2009).

Despite our quantum algorithm having a running time that achieves quadratic improvement in the dimension $d$ of the weight vector, its dependence on the number of time steps $T$ increases. One natural question would be to ask if the trade-off between $T$ and $d$ can be avoided. Besides that, it would be interesting to explore how other variants of gradient descent such as mirror descent or stochastic gradient descent, combined with different "feature selection" techniques to obtain sparse solutions can contribute to an improvement in the regret bound. Considering that we have a unitary that computes entries of the weight vector that is updated via truncated gradient descent , one could consider potential applications of this unitary, for example in reinforcement learning (Mahadevan and Liu, 2012). On the other hand, one could explore possible applications of quantum algorithms in obtaining sparse solutions in the online learning setting as there has not been any work done in this regime. Instead of analyzing the (static) regret, one could consider studying the *dynamic* regret of the online algorithm which can be useful in scenarios where the optimal solution keeps changing in evolving environments (Besbes et al., 2015; Jadbabaie et al., 2015; Mokhtari et al., 2016; Yang et al., 2016; Zhang et al., 2017; 2018; Zhao et al., 2020b).

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

## A  APPENDIX / SUPPLEMENTAL MATERIAL

### A.1  ALGORITHMS

---

**Algorithm 2** Gradient descent

---

**Require:** Loss function $L$, total time steps $T$, initial point $w^{(1)}$, learning rates $\eta^{(1)}, \cdots, \eta^{(T)}$
**Ensure:** $w^{(T+1)}$
1: **for** $t = 1$ **to** $T$ **do**
2:    Let $w^{(t+1)} = w^{(t)} - \eta^{(t)} \nabla L(w^{(t)})$.
3: **end for**

---

---

**Algorithm 3** Truncated gradient descent

---

**Require:** Convex loss function $L$, total time steps $T$, initial point $w^{(1)}$, learning rates $\eta^{(1)}, \cdots, \eta^{(T)}$, threshold $\theta$, constant $K$, shrinkage parameters $\alpha^{(1)}, \cdots, \alpha^{(T)}$.
1: **for** $t = 1$ **to** $T$ **do**
2:    Let $w'^{(t+1)} = w^{(t)} + \eta^{(t)} \nabla L(w^{(t)})$.
3:    **for** $j = 1, \cdots, d$ **do**
4:      **if** $0 \leq w'^{(t+1)} \leq \theta$ and $\frac{t}{K}$ is an integer **then**
5:        $w_j^{(t+1)} = \max\left\{ w_j'^{(t+1)} - \alpha^{(t)}, 0 \right\}$.
6:      **else if** $-\theta \leq w'^{(t+1)} \leq 0$ and $\frac{t}{K}$ is an integer **then**
7:        $w_j^{(t+1)} = \min\left\{ w_j'^{(t+1)} + \alpha^{(t)}, 0 \right\}$.
8:      **else**
9:        $w_j^{(t+1)} = w_j'^{(t+1)}$.
10:     **end if**
11:   **end for**
12: **end for**
**Ensure:** $w^{(T+1)}$.

---

---

**Algorithm 4** Online sparse learning algorithm with truncated gradient descent

---

**Require:** Threshold $\theta > 0$, gravity sequence $\{g^{(1)}, \cdots, g^{(T)}\} \leq g_{\max}$, learning rate $\eta \in (0, 1)$, example oracle $\mathcal{O}$.
1: Initialize $w^{(1)} = (0, \cdots, 0) \in \mathbb{R}^d$.
2: **for** $t = 1$ **to** $T$ **do**
3:    Receive unlabeled example $x^{(t)} = \left( x_1^{(t)}, \cdots, x_d^{(t)} \right) \in \mathbb{R}^d$ form example oracle $\mathcal{O}$.
4:    Compute the prediction $\hat{y}^{(t)} = \sum_{j=1}^{d} w_j^{(t)} x_j^{(t)}$.
5:    Receive the true label $y^{(t)}$ from example oracle $\mathcal{O}$.
6:    **for** $j = 1$ **to** $d$ **do**
7:      $w_j'^{(t+1)} \leftarrow w_j^{(t)} - \eta \nabla L\left( w^{(t)} \right)$.
8:    **end for**
9:    **for** $j = 1$ **to** $d$ **do**
10:     **if** $0 \leq w_j^{(t)} \leq \theta$ **then**
11:       $w_j^{(t)} \leftarrow \max\left\{ w_j'^{(t)} - g^{(t)}\eta, 0 \right\}$
12:     **else if** $-\theta \leq w_j^{(t)} \leq 0$ **then**
13:       $w_j^{(t)} \leftarrow \min\left\{ w_j'^{(t)} + g^{(t)}\eta, 0 \right\}$
14:     **else**
15:       $w_j^{(t)} \leftarrow w_j^{(t)}$
16:     **end if**
17:   **end for**
18: **end for**

---

## A.2 PROOF OF REGRET BOUND

**Fact 2** (Theorem 3.1, (Langford et al., 2009))**.** *Suppose that Assumption 1 holds. Then, for all* $u \in \mathbb{R}^d$*, Algorithm 4 achieves*

$$\frac{1 - 0.5A\eta}{T} \sum_{t=1}^{T} \left[ L\left(w^{(t)}, x^{(t)}, y^{(t)}\right) + \frac{g^{(t)}}{1 - 0.5A\eta} \left\| w^{(t+1)} \cdot I\left(\left| w^{(t+1)} \right| \leq \theta\right) \right\|_1 \right] \quad (2)$$

$$\leq \quad \frac{\eta}{2}B + \frac{\|u\|_2^2}{2\eta T} + \frac{1}{T} \sum_{t=1}^{T} \left[ L\left(u, x^{(t)}, y^{(t)}\right) + g^{(t)} \left\| u \cdot I\left(\left| w^{(t+1)} \right| \leq \theta\right) \right\|_1 \right] \quad (3)$$

## A.3 PROOF OF LEMMA 1

Start with the state $|\alpha\rangle \, |-\theta\rangle \, |0\rangle \, |\theta\rangle \, |x\rangle \, |0\rangle \, |0\rangle \, |0\rangle \, |0\rangle$. Query the Between oracle on the third to sixth registers. Then,

(i) Condition on the sixth register being $|1\rangle$, flip the seventh register to flag that $0 \leq x \leq \theta$, and do

$$|\alpha\rangle \, |-\theta\rangle \, |0\rangle \, |\theta\rangle \, |x\rangle \, |1\rangle \, |1\rangle \, |0\rangle \, |0\rangle \to |\alpha\rangle \, |-\theta\rangle \, |0\rangle \, |\theta\rangle \, |x\rangle \, |1\rangle \, |1\rangle \, |x - \alpha\rangle \, |0\rangle. \quad (4)$$

Next, query the Minmax oracle on the seventh and eighth register to get

$$|\alpha\rangle \, |-\theta\rangle \, |0\rangle \, |\theta\rangle \, |x\rangle \, |1\rangle \, |1\rangle \, |x - \alpha\rangle \, |0\rangle \to |\alpha\rangle \, |-\theta\rangle \, |0\rangle \, |\theta\rangle \, |x\rangle \, |1\rangle \, |1\rangle \, |x - \alpha\rangle \, |\max(x - \alpha, 0)\rangle. \quad (5)$$

Lastly, swap the fifth and the last registers and uncompute intermediate registers.

(ii) Condition on the sixth register being $|0\rangle$, query the between oracle on the second, third, fifth and sixth registers. Then,

  (i) condition on the sixth being $|1\rangle$, do

  $$|\alpha\rangle \, |-\theta\rangle \, |0\rangle \, |\theta\rangle \, |x\rangle \, |1\rangle \, |0\rangle \, |0\rangle \, |0\rangle \to |\alpha\rangle \, |-\theta\rangle \, |0\rangle \, |\theta\rangle \, |x\rangle \, |1\rangle \, |0\rangle \, |x + \alpha\rangle \, |0\rangle. \quad (6)$$

  Next, query the Minmax oracle on the eighth and ninth registers to get

  $$|\alpha\rangle \, |-\theta\rangle \, |0\rangle \, |\theta\rangle \, |x\rangle \, |1\rangle \, |0\rangle \, |x + \alpha\rangle \, |0\rangle \to |\alpha\rangle \, |-\theta\rangle \, |0\rangle \, |\theta\rangle \, |x\rangle \, |1\rangle \, |0\rangle \, |x + \alpha\rangle \, |\min(x + \alpha, 0)\rangle. \quad (7)$$

  Lastly, swap the sixth and the last registers and uncompute intermediate registers.

  (ii) Condition on the sixth register being $|0\rangle$, do nothing.

Since we assume the use of the quantum arithmetic model, we hence obtain a running time of $\tilde{O}(1)$.

## A.4 PROOF OF LEMMA 2

(i) Using the query access, create the circuit to prepare the state $\frac{1}{\sqrt{d}} \sum_{j=1}^{d} |j\rangle \, |u_j\rangle \, |0\rangle$. Use quantum maximum finding in Durr and Hoyer (1996) to find

$$\|u\|_\infty := \max_{j \in [d]} |u_j| \quad (8)$$

with success probability $1 - \delta/2$. Apply a controlled-rotation to the state obtains

$$\frac{1}{\sqrt{d}} \sum_{j=1}^{d} |j\rangle \, |u_j\rangle \left( \sqrt{\frac{|u_j|}{\|u\|_\infty}} \, |0\rangle + \sqrt{1 - \frac{|u_j|}{\|u\|_\infty}} \, |1\rangle \right). \quad (9)$$

Let $U_u$ be the unitary that prepares the state in Eq. (9). Define new unitaries $U = U_u(I - 2\, |\bar{0}\rangle \, \langle\bar{0}|)U_u^\dagger$ and $V = I - I \otimes |0\rangle \, \langle 0|$. Fact 1 allows us to obtain an estimate $\tilde{a}$ of $a = \frac{\|u\|_1}{d\|u\|_\infty}$ such that $|\mathbb{E}[\tilde{a}] - a| \leq \frac{\epsilon_0^2}{32}a^2$ and $\text{Var}(\tilde{a}) \leq \frac{91a}{K^2} + \frac{\epsilon_0^2}{32}a^2$, restoring the initial state with success probability at least $1 - \frac{\epsilon_0^2}{32}a^2$, using $O\left(K \log\log K \log(K/\epsilon_0)\right)$ expected number of

applications of $U$ and $V$. Setting $K > \frac{8}{\epsilon_0}\sqrt{\frac{91}{a}}$ via exponential search without knowledge of $a$, we have

$$\mathbb{P}\left[|\tilde{a} - \mathbb{E}[\tilde{a}]| \geq \frac{\epsilon_0}{2}a\right] \leq \frac{4}{\epsilon_0^2 a^2}\left(\frac{91a}{K^2} + \frac{\epsilon_0^2 a^2}{32}\right) \leq \frac{4}{\epsilon_0^2 a^2}\left(\frac{\epsilon_0^2 a^2}{64} + \frac{\epsilon_0^2 a^2}{32}\right) \leq \frac{1}{16} + \frac{1}{8} \leq \frac{1}{4}.$$

by Chebyshev's inequality. The success probability $3/4$ is boosted with $O(\log\frac{1}{\delta})$ repetitions to $1 - \delta/2$ via the median of means technique. Hence,

$$|\tilde{a} - a| \leq |\tilde{a} - \mathbb{E}[\tilde{a}]| + |\mathbb{E}[\tilde{a}] - a| \leq \epsilon_0 a/2 + \epsilon_0 a/2 = \epsilon_0 a$$

with success probability at least $1 - 4\delta$. Hence the quantity $\tilde{\Gamma} := \tilde{a}$ is an estimate

$$\left|\tilde{\Gamma} - \frac{\|u\|_1}{d\|u\|_\infty}\right| = |\tilde{a} - a| \leq \epsilon_0 a. \tag{10}$$

This brings the total time complexity to

$$O\left(\frac{1}{\epsilon_0}\sqrt{\frac{d\|u\|_\infty}{\|u\|_1}}\log\log\left(\frac{1}{\epsilon_0}\sqrt{\frac{d\|u\|_\infty}{\|u\|_1}}\right)\log\left(\frac{1}{\epsilon_0^2}\left(\frac{d\|u\|_\infty}{\|u\|_1}\right)^{3/2}\right)\log\frac{1}{\delta}\right)$$

in expectation. While Cornelissen and Hamoudi (2023) proves a result in expected time, we use the probabilistic result obtained from Markov's inequality and repetition at a cost of another factor of $O\left(\log\frac{1}{\delta}\right)$.

We note that the additive version of norm estimation can be easily specialized by setting $K > \frac{8}{\epsilon_0}\sqrt{91a}$ and has the same time and gate complexity.

(ii) Note that the state in Eq. (9) can be rewritten as

$$\sqrt{\frac{\|u\|_1}{\|u\|_\infty d}}\sum_{j=1}^{d}|j\rangle|u_j\rangle\left(\sqrt{\frac{|u_j|}{\|u\|_1}}|0\rangle + \sqrt{1 - \frac{|u_j|}{\|u\|_1}}|1\rangle\right). \tag{11}$$

Amplify the $|0\rangle$ part via amplitude amplification (Brassard et al., 2002) to obtain $|u\rangle = \sum_{j=1}^{d}\sqrt{\frac{|u_j|}{\|u\|_1}}|j\rangle$ with success probability $1 - \delta$ using $O\left(\sqrt{\frac{\|u\|_\infty d}{\|u\|_1}}\log\frac{1}{\delta}\right) \subseteq O\left(\sqrt{d}\log\frac{1}{\delta}\right)$ calls to $U_u$ and $\tilde{O}\left(\sqrt{d}\log\frac{1}{\zeta}\log\frac{1}{\delta}\right)$ gates. For all $j \in [d]$, we have $\tilde{p}_j = \frac{u_j}{\tilde{\Gamma}}$. Also, note that $\|u\|_1 - \tilde{\Gamma} \leq |\|u\|_1 - \tilde{\Gamma}| \leq \zeta\|u\|_1$, and hence $\frac{1}{\tilde{\Gamma}} \geq \frac{1}{\|u\|_1(1-\zeta)}$. Therefore, $\|\tilde{p} - \frac{u}{\|u\|_1}\|_1 = \sum_{j=1}^{d}|\frac{u_j}{\tilde{\Gamma}} - \frac{u_j}{\|u\|_1}| = \sum_{j=1}^{d}\left|\frac{u_j\|u\|_1 - u_j\tilde{\Gamma}}{\tilde{\Gamma}\|u\|_1}\right| = \sum_{j=1}^{d}\left|\frac{u_j(\|u\|_1 - \tilde{\Gamma})}{\tilde{\Gamma}\|u\|_1}\right| \leq \sum_{j=1}^{d}\left|\frac{u_j\cdot\zeta\|u\|_1}{\|u\|_1\cdot(1-\zeta)\|u\|_1}\right|$

$= \sum_{j=1}^{d}\left|\frac{u_j}{\|u\|_1}\cdot\frac{\zeta}{(1-\zeta)}\right| = \frac{\zeta}{(1-\zeta)}\sum_{j=1}^{d}\left|\frac{u_j}{\|u\|_1}\right| = \frac{\zeta}{(1-\zeta)} \leq 2\zeta.$

## A.5 PROOF OF LEMMA 3

Define vectors $u^+$ and $u^-$ as follows

$$u_i^+ = \begin{cases} u_i, & \text{if } \text{sign}(u_i) = 1 \\ 0, & \text{otherwise} \end{cases}, \qquad u_i^- = \begin{cases} 0, & \text{if } \text{sign}(u_i) = 1 \\ -u_i, & \text{otherwise} \end{cases} \tag{12}$$

Notice that $u = u^+ - u^-$. Define $v^+$ and $v^-$ in a similar way. Then

$$u \cdot v = u^+ \cdot v^+ + u^- \cdot v^- - u^- \cdot v^+ - u^+ \cdot v^- \tag{13}$$

Let $z^+$ and $z^-$ be vectors such that $z_i^+ = u_i^+ \cdot v_i^+ + u_i^- \cdot v_i^-$ and $z_i^- = u_i^+ \cdot v_i^- + u_i^- \cdot v_i^+$. Then, observe that

$$u \cdot v = \|z^+\|_1 - \|z^-\|_1 \tag{14}$$

Next, determine $z_{\max}^+ := \max_{j\in[d]}|z_j^+|$ using quantum maximum finding with success probability $1 - \delta/4$ using $O\left(\sqrt{d}\log\frac{1}{\delta}\right)$ queries and $\tilde{O}\left(\sqrt{d}\log\frac{1}{\delta}\right)$ quantum gates (Durr and Hoyer, 1996).

If $z_{\max}^+ = 0$ up to sufficient numerical accuracy, then $\widetilde{IP} = 0$ and we are done. Otherwise, use Lemma 2(i) to obtain an estimate $\tilde{\Gamma}_{z^+}$ of $\left\| \frac{z^+}{z_{\max}^+} \right\|_1$ such that

$$\left| \left\| \frac{z^+}{z_{\max}^+} \right\|_1 - \tilde{\Gamma}_{z^+} \right| \leq \frac{\epsilon^+}{2} \left\| \frac{z^+}{z_{\max}^+} \right\|_1 \tag{15}$$

with success probability at least $1 - \delta/4$ with $\tilde{O}\left( \frac{1}{\epsilon^+} \sqrt{\frac{dz_{\max}^+}{\|z^+\|_1}} \log \frac{1}{\delta} \right) \subseteq \tilde{O}\left( \frac{\sqrt{d}}{\epsilon^+} \log \frac{1}{\delta} \right)$ queries and $\tilde{O}\left( \frac{\sqrt{d}}{\epsilon^+} \log \frac{1}{\delta} \right)$ quantum gates.

Similarly for the case of $z^-$, find $z_{\max}^- := \max_{j \in [d]} |z^-|$ using quantum maximum finding with success probability $1 - \delta/4$ using $O\left( \sqrt{d} \log \frac{1}{\delta} \right)$ queries and $\tilde{O}\left( \sqrt{d} \log \frac{1}{\delta} \right)$ quantum gates (Durr and Hoyer, 1996). If $z_{\max}^- = 0$ up to sufficient numerical accuracy, then $\widetilde{IP} = 0$ and we are done. Otherwise, use Lemma 2(i) to obtain an estimate $\tilde{\Gamma}_{z^-}$ of $\left\| \frac{z^-}{z_{\max}^-} \right\|_1$ such that

$$\left| \left\| \frac{z^-}{z_{\max}^-} \right\|_1 - \tilde{\Gamma}_{z^-} \right| \leq \frac{\epsilon^-}{2} \left\| \frac{z^-}{z_{\max}^-} \right\|_1 \tag{16}$$

with success probability at least $1 - \delta/4$ with $\tilde{O}\left( \frac{1}{\epsilon^-} \sqrt{\frac{dz_{\max}^+}{\|z^+\|_1}} \log \frac{1}{\delta} \right) \subseteq O\left( \frac{\sqrt{d}}{\epsilon^-} \log \frac{1}{\delta} \right)$ queries and $\tilde{O}\left( \frac{\sqrt{d}}{\epsilon^-} \log \frac{1}{\delta} \right)$ quantum gates.

Set $\widetilde{IP} = z_{\max}^+ \tilde{\Gamma}_{z^+} - z_{\max}^- \tilde{\Gamma}_{z^-}$. Then

$$\left| u \cdot v - \widetilde{IP} \right| = \left| \|z^+\|_1 - \|z^-\|_1 - \left( z_{\max}^+ \tilde{\Gamma}^+ - z_{\max}^- \tilde{\Gamma}^- \right) \right|$$

$$\text{(triangle ineq.)} \leq \left| \|z^+\|_1 - z_{\max}^+ \tilde{\Gamma}^+ \right| + \left| \|z^-\|_1 - z_{\max}^- \tilde{\Gamma}^- \right|$$

$$\text{(Lemma 2(i))} \leq \frac{\epsilon^+}{2} \|z^+\|_1 + \frac{\epsilon^-}{2} \|z^-\|_1$$

$$= \frac{\epsilon^+}{2} \left( \|u^+ \cdot v^+\|_1 + \|u^- \cdot v^-\|_1 \right) + \frac{\epsilon^-}{2} \left( u^+ \cdot v^- + u^- \cdot v^+ \right)$$

$$\text{(Hölder's ineq.)} \leq \frac{\epsilon^+}{2} \left( \|u^+\|_\infty \|v^+\|_1 + \|u^-\|_\infty \|v^-\|_1 \right) + \frac{\epsilon^-}{2} \left( \|u^+\|_\infty \|v^-\|_1 + \|u^-\|_\infty \|v^+\|_1 \right)$$

$$\leq \frac{\epsilon^+}{2} \|u\|_\infty \left( \|v^+\|_1 + \|v^-\|_1 \right) + \frac{\epsilon^-}{2} \|u\|_\infty \left( \|v^+\|_1 + \|v^-\|_1 \right)$$

$$= \left( \frac{\epsilon^+}{2} + \frac{\epsilon^-}{2} \right) \|u\|_\infty \|v\|_1$$

Setting $\epsilon^+ = \epsilon^- = \frac{\epsilon}{\|u\|_\infty \|v\|_1}$ yields the desired result. The total time complexity is $\tilde{O}\left( \frac{\|u\|_\infty \|v\|_1 \sqrt{d}}{\epsilon} \log \frac{1}{\delta} \right)$.

## A.6 PROOF OF LEMMA 4

With a computational register of $O(T)$ ancilla qubits for the examples, perform

$$|j\rangle |\bar{0}\rangle$$
$$\to |j\rangle |x_j^{(1)}\rangle \cdots |x_j^{(t)}\rangle |\bar{0}\rangle$$
$$\to |j\rangle |x_j^{(1)}\rangle \cdots |x_j^{(t)}\rangle |w_j'^{(1)}\rangle |\bar{0}\rangle \tag{17}$$

(i) For $t \in [T]$, let $w_j'^{(t)} = w_j^{(t-1)} + 2\eta \left( y^{(t-1)} - \tilde{y}^{(t-1)} \right) x_j^{(t-1)}$ and $w_j^{(t)} = \mathcal{T}\left( w_j'^{(t)}, g^{(t)}\eta, \theta \right)$. Then, from Eq. (17), perform the following operations:

$$\xrightarrow{U_{\mathcal{T}, g^{(1)}\eta, \theta}} |j\rangle |x_j^{(1)}\rangle \cdots |x_j^{(t)}\rangle |w_j'^{(1)}\rangle |w_j^{(1)}\rangle |\bar{0}\rangle$$

$$\vdots$$

$$\rightarrow |j\rangle |x_j^{(1)}\rangle \cdots |x_j^{(t)}\rangle |w_j'^{(1)}\rangle |w_j^{(1)}\rangle \cdots |w_j'^{(t)}\rangle |\bar{0}\rangle$$

$$\xrightarrow{U_{\mathcal{T},g^{(t)}\eta,\theta}} |j\rangle |x_j^{(1)}\rangle \cdots |x_j^{(t)}\rangle |w_j'^{(1)}\rangle |w_j^{(1)}\rangle \cdots |w_j'^{(t)}\rangle |w_j^{(t)}\rangle$$

(ii) For $t \in [T]$, let $w_j'^{(t)} = w_j^{(t-1)} + 2\eta \frac{x_j^{(t)} y^{(t)} e^{-y^{(t)} \tilde{y}^{(t)}}}{1 + e^{-y^{(t)} \tilde{y}^{(t)}}}$ and $w_j^{(t)} = \mathcal{T}\left(w_j'^{(t)}, g^{(t)}\eta, \theta\right)$. Then, from Eq. (17), perform the following operations:

$$\xrightarrow{U_{\mathcal{T},g^{(1)}\eta,\theta}} |j\rangle |x_j^{(1)}\rangle \cdots |x_j^{(t)}\rangle |w_j'^{(1)}\rangle |w_j^{(1)}\rangle |\bar{0}\rangle$$

$$\vdots$$

$$\rightarrow |j\rangle |x_j^{(1)}\rangle \cdots |x_j^{(t)}\rangle |w_j'^{(1)}\rangle |w_j^{(1)}\rangle \cdots |w_j'^{(t)}\rangle |\bar{0}\rangle$$

$$\xrightarrow{U_{\mathcal{T},g^{(t)}\eta,\theta}} |j\rangle |x_j^{(1)}\rangle \cdots |x_j^{(t)}\rangle |w_j'^{(1)}\rangle |w_j^{(1)}\rangle \cdots |w_j'^{(t)}\rangle |w_j^{(t)}\rangle$$

(iii) For $t \in [T]$, let

$$w_j'^{(t)} = \begin{cases} w_j^{(t-1)} + \eta y^{(t-1)} x_j^{(t-1)}, & \text{if } y^{(t-1)} \tilde{y}^{(t-1)} < 1 \\ w_j^{(t-1)}, & \text{otherwise} \end{cases} \tag{18}$$

and $w_j^{(t)} = \mathcal{T}\left(w_j'^{(t)}, g^{(t)}\eta, \theta\right)$. Then, from Eq. (17), perform the following operations:

$$|j\rangle |\bar{0}\rangle$$

$$\rightarrow |j\rangle |x_j^{(1)}\rangle \cdots |x_j^{(t)}\rangle |\bar{0}\rangle$$

$$\rightarrow |j\rangle |x_j^{(1)}\rangle \cdots |x_j^{(t)}\rangle |w_j'^{(1)}\rangle |\bar{0}\rangle$$

$$\xrightarrow{U_{\mathcal{T},g^{(1)}\eta,\theta}} |j\rangle |x_j^{(1)}\rangle \cdots |x_j^{(t)}\rangle |w_j'^{(1)}\rangle |w_j^{(1)}\rangle |\bar{0}\rangle$$

$$\vdots$$

$$\rightarrow |j\rangle |x_j^{(1)}\rangle \cdots |x_j^{(t)}\rangle |w_j'^{(1)}\rangle |w_j^{(1)}\rangle \cdots |w_j'^{(t)}\rangle |\bar{0}\rangle$$

$$\xrightarrow{U_{\mathcal{T},g^{(t)}\eta,\theta}} |j\rangle |x_j^{(1)}\rangle \cdots |x_j^{(t)}\rangle |w_j'^{(1)}\rangle |w_j^{(1)}\rangle \cdots |w_j'^{(t)}\rangle |w_j^{(t)}\rangle$$

with sufficient accuracy using the oracles and quantum circuits for arithmetic operations. Uncomputing the intermediate registers will yield the desired result.

**Remark 1.** *For the addition of two integers each encoded in $k$-bit binary strings, the quantum circuit for addition has a Toffoli count of $2k - 1$ and Toffoli depth of $k$ (Takahashi et al., 2009). On the other hand, the quantum circuit for integer division has a Toffoli count of $14k^2 + 7k + 7$ and a Toffoli depth of $10k + 13$ while subtraction has a Toffoli count of $O(k)$ and a Toffoli depth of $O(1)$ (Thapliyal et al., 2019). Circuits for floating-point addition are generated using synthesis tools and can be hand-optimized as shown in Haener et al. (2018). Furthermore, circuit sizes for floating-point division have been computed numerically by Gayathri et al. (2021), which improves upon the work of Nguyen and Van Meter (2014) and Nachtigal et al. (2010). In the above computation, we perform $T$ number of divisions and $T$ number of additions, which leads to a circuit of approximately $O(Tk^2)$ size and $O(Tk)$ depth. Since we assume the use of the quantum arithmetic model, we hence obtain the $\tilde{O}(T + \log n)$ gate complexity.*

## A.7 Proof of Theorem 1

We start by considering the following expression for the regret bound:

$$\frac{1}{T}\sum_{t=1}^{T}\ln\left(1+e^{-y^{(t)}\tilde{y}^{(t)}}\right)+\frac{1}{T}\sum_{t=1}^{T}g^{(t)}\tilde{q}^{(t+1)}$$

$$-\frac{1}{T}\sum_{t=1}^{T}\ln\left(1+e^{-y^{(t)}u^{\mathsf{T}}x^{(t)}}\right)-\frac{1}{T}\sum_{t=1}^{T}g^{(t)}\left\|u\cdot I\left(\left|w^{(t+1)}\right|\right)\right\|_{1}$$

Next, we simplify this by separating terms related to $\hat{y}^{(t)}$ and $\tilde{y}^{(t)}$ and apply Fact 2:

$$\leq\frac{1}{T}\sum_{t=1}^{T}\ln\left(1+e^{-y^{(t)}\tilde{y}^{(t)}}\right)+\frac{1}{T}\sum_{t=1}^{T}g^{(t)}\tilde{q}^{(t+1)}$$

$$-\frac{1}{T}\sum_{t=1}^{T}\ln\left(1+e^{-y^{(t)}\hat{y}^{(t)}}\right)-\frac{1}{T}\sum_{t=1}^{T}g^{(t)}q^{(t+1)}+\frac{\eta C^2}{2}+\frac{\|u\|_2^2}{2\eta T}$$

$$\leq\frac{1}{T}\sum_{t=1}^{T}\left|\ln\left(1+e^{-y^{(t)}\tilde{y}^{(t)}}\right)-\ln\left(1+e^{-y^{(t)}\hat{y}^{(t)}}\right)\right|$$

$$+\frac{1}{T}\sum_{t=1}^{T}g^{(t)}\left|\tilde{q}^{(t+1)}-q^{(t+1)}\right|+\frac{\eta C^2}{2}+\frac{\|u\|_2^2}{2\eta T}$$

Using additive variant of Lemma 2(i), we further simplify it:

$$\leq\frac{1}{T}\sum_{t=1}^{T}\left|\ln\left(1+e^{-y^{(t)}\tilde{y}^{(t)}}\right)-\ln\left(1+e^{-y^{(t)}\hat{y}^{(t)}}\right)\right|+\frac{\epsilon_{\mathrm{norm}}}{T}\sum_{t=1}^{T}g^{(t)}+\frac{\eta C^2}{2}+\frac{\|u\|_2^2}{2\eta T}$$

Bounding $g^{(t)}$ by $g_{\mathrm{max}}$ gives us

$$\leq\frac{1}{T}\sum_{t=1}^{T}\left|\ln\left(1+e^{-y^{(t)}\tilde{y}^{(t)}}\right)-\ln\left(1+e^{-y^{(t)}\hat{y}^{(t)}}\right)\right|+g_{\mathrm{max}}\epsilon_{\mathrm{norm}}+\frac{\eta C^2}{2}+\frac{\|u\|_2^2}{2\eta T}$$

Finally, by applying the inequality $\ln(1+x)\leq x$ for $x>-1$ and leveraging Lipschitz continuity, as well as Lemma 3, we obtain:

$$\leq\frac{1}{T}\sum_{t=1}^{T}e\epsilon_{\mathrm{IP}}+g_{\mathrm{max}}\epsilon_{\mathrm{norm}}+\frac{\eta C^2}{2}+\frac{\|u\|_2^2}{2\eta T}$$

$$=e\epsilon_{\mathrm{IP}}+g_{\mathrm{max}}\epsilon_{\mathrm{norm}}+\frac{\eta C^2}{2}+\frac{\|u\|_2^2}{2\eta T}.$$

Setting $\epsilon_{\mathrm{IP}}=\frac{1}{4\eta T}$, $\epsilon_{\mathrm{norm}}=\frac{1}{2\eta T}$ and $\eta=\frac{1}{C^2\sqrt{T}}$, the desired regret bound is achieved with success probability $1-\delta$ by the union bound when quantum inner product and norm estimation each succeeds with probability $1-\delta/2$.

Finally, we compute the total time complexity of the algorithm. The algorithm uses the following subroutines, each with the correspoinding running time:

$T$ (quantum inner product estimation + quantum norm estimation + quantum state preparation)

$$\subseteq O\left(T\left(\frac{T\sqrt{d}}{\epsilon_{IP}}\log\frac{T}{\delta}+\frac{T\sqrt{d}}{\epsilon_{\mathrm{norm}}}\log\frac{T}{\delta}+T\sqrt{d}\log\frac{T}{\delta}\right)\right)\subseteq O\left(T^{5/2}\sqrt{d}\log\frac{T}{\delta}\right).$$

## A.8 Proof of Theorem 2

We begin by noting the expression:

$$\frac{1}{T}\sum_{t=1}^{T}\left(1-y^{(t)}\tilde{y}^{(t)}\right)^{+}+\frac{1}{T}\sum_{t=1}^{T}g^{(t)}\tilde{q}^{(t+1)}$$

$$-\frac{1}{T}\sum_{t=1}^{T}\left(1-y^{(t)}u^{\mathsf{T}}x^{(t)}\right)^{+}-\frac{1}{T}\sum_{t=1}^{T}g^{(t)}\left\|u\cdot I\left(\left|w^{(t+1)}\right|\right)\right\|_{1}$$

Next, we separate the terms for terms for $\hat{y}^{(t)}$ and $\tilde{y}^{(t)}$ and apply Fact 2:

$$\leq \frac{1}{T}\sum_{t=1}^{T}\left(1-y^{(t)}\tilde{y}^{(t)}\right)^{+}+\frac{1}{T}\sum_{t=1}^{T}g^{(t)}\tilde{q}^{(t+1)}$$

$$-\frac{1}{T}\sum_{t=1}^{T}\left(1-y^{(t)}\hat{y}^{(t)}\right)^{+}-\frac{1}{T}\sum_{t=1}^{T}g^{(t)}q^{(t+1)}+\frac{C^2\|u\|_2^2}{2\sqrt{T}}+\frac{1}{2\sqrt{T}}$$

$$\leq \frac{1}{T}\sum_{t=1}^{T}\left|\left(1-y^{(t)}\tilde{y}^{(t)}\right)^{+}-\left(1-y^{(t)}\hat{y}^{(t)}\right)^{+}\right|$$

$$+\frac{1}{T}\sum_{t=1}^{T}g^{(t)}\left|\tilde{q}^{(t+1)}-q^{(t+1)}\right|+\frac{C^2\|u\|_2^2}{2\sqrt{T}}+\frac{1}{2\sqrt{T}}$$

Using additive variant of Lemma 2(i) and $g_{\max}=\max_{t\in[T]}g^{(t)}$, we obtain:

$$\leq \frac{1}{T}\sum_{t=1}^{T}\left|\left(1-y^{(t)}\tilde{y}^{(t)}\right)^{+}-\left(1-y^{(t)}\hat{y}^{(t)}\right)^{+}\right|+\epsilon_{\text{norm}}g_{\max}+\frac{C^2\|u\|_2^2}{2\sqrt{T}}+\frac{1}{2\sqrt{T}}$$

We finally obtain the bound by using Lipschitz continuity and Lemma 3:

$$\leq |\hat{y}^{(t)}-\tilde{y}^{(t)}|+\epsilon_{\text{norm}}g_{\max}+\frac{C^2\|u\|_2^2}{2\sqrt{T}}+\frac{1}{2\sqrt{T}}$$

$$\leq \epsilon_{\text{IP}}+\epsilon_{\text{norm}}g_{\max}+\frac{C^2\|u\|_2^2}{2\sqrt{T}}+\frac{1}{2\sqrt{T}}$$

$$\leq \frac{2+C^2\left(g_{\max}+\|u\|_2^2\right)}{2\sqrt{T}}$$

when setting $\epsilon_{\text{IP}}=\frac{1}{2\sqrt{T}}$, $\epsilon_{\text{norm}}=\frac{1}{2\eta T}$ and $\eta=\frac{1}{C^2\sqrt{T}}$. This succeeds with probability $1-\delta$ by the union bound when quantum inner product and norm estimation each succeeds with probability $1-\delta/2$.

The time complexity analysis is the same as that of Theorem 1.

### A.9 PROOF OF THEOREM 3

Note that

$$\frac{1-2C^2\eta}{T}\sum_{t=1}^{T}\left(\tilde{y}^{(t)}-y^{(t)}\right)^2+\frac{1}{T}\sum_{t=1}^{T}g^{(t)}\tilde{q}^{(t+1)}$$

$$-\frac{1}{T}\sum_{t=1}^{T}\left(u^{\mathsf{T}}x^{(t)}-y^{(t)}\right)^2-\frac{1}{T}\sum_{t=1}^{T}g^{(t)}\left\|u\cdot I\left(\left|w^{(t+1)}\right|\leq\theta\right)\right\|_1$$

At this point, we apply Fact 2 to obtain the following bound:

$$\leq \frac{1-2C^2\eta}{T}\sum_{t=1}^{T}\left(\tilde{y}^{(t)}-y^{(t)}\right)^2+\frac{1}{T}\sum_{t=1}^{T}g^{(t)}\tilde{q}^{(t+1)}$$

$$-\frac{1-2C^2\eta}{T}\sum_{t=1}^{T}\left(\hat{y}^{(t)}-y^{(t)}\right)^2-\frac{1}{T}\sum_{t=1}^{T}g^{(t)}\left\|w^{(t+1)}\cdot I\left(\left|w^{(t+1)}\right|\leq\theta\right)\right\|_1+\frac{\|u\|_2^2}{2\eta T}$$

Next, we separate the terms to focus on the error:

$$\leq \frac{1-2C^2\eta}{T}\sum_{t=1}^{T}\left|\left(\tilde{y}^{(t)}-y^{(t)}\right)^2-\left(\hat{y}^{(t)}-y^{(t)}\right)^2\right|+\frac{1}{T}\sum_{t=1}^{T}g^{(t)}\left|\tilde{q}^{(t+1)}-q^{(t+1)}\right|+\frac{\|u\|_2^2}{2\eta T}$$

Use Lipschitz continuity to further bound it:

$$\leq \frac{(1 - 2C^2\eta)\epsilon_{\text{IP}}}{T} \sum_{t=1}^{T} 2 \left| \hat{y}^{(t)} - y^{(t)} \right| \left\| x^{(t)} \right\|_2 + \frac{1}{T} \sum_{t=1}^{T} g^{(t)} \left| \tilde{q}^{(t+1)} - q^{(t+1)} \right| + \frac{\|u\|_2^2}{2\eta T}$$

By additive variant of Lemma 2(i), now we have:

$$\leq \frac{(1 - 2C^2\eta)\epsilon_{\text{IP}}}{T} \sum_{t=1}^{T} 2 \left| \hat{y}^{(t)} - y^{(t)} \right| \left\| x^{(t)} \right\|_2 + \frac{1}{T} \sum_{t=1}^{T} g^{(t)} \epsilon_{\text{norm}} + \frac{\|u\|_1}{2\eta T}$$

Applying assumption 1(iii), we can simplify it:

$$\leq \frac{2C\epsilon_{\text{IP}}(1 - 2C^2\eta)}{T} \sum_{t=1}^{T} \left| \hat{y}^{(t)} - y^{(t)} \right| + \frac{1}{T} \sum_{t=1}^{T} g^{(t)} \epsilon_{\text{norm}} + \frac{\|u\|_2^2}{2\eta T}$$

Finally, we apply $g_{\max} = \max_{t \in [T]} g^{(t)}$ and assumption 2 to obtain:

$$\leq \frac{2C\epsilon_{\text{IP}}(1 - 2C^2\eta)}{T} \sum_{t=1}^{T} \left| \hat{y}^{(t)} - y^{(t)} \right| + \frac{\epsilon_{\text{norm}}}{T} \sum_{t=1}^{T} g_{\max} + \frac{\|u\|_2^2}{2\eta T}$$

$$\leq 2C\epsilon_{\text{IP}}(1 - 2C^2\eta)D + \epsilon_{\text{norm}}g_{\max} + \frac{\|u\|_2^2}{2\eta T}$$

$$\leq 2CD\epsilon_{\text{IP}} + \epsilon_{\text{norm}}g_{\max} + \frac{\|u\|_2^2}{2\eta T}$$

with success probability $1 - \delta$ by the union bound when quantum inner product and norm estimation each succeeds with probability $1 - \delta/2$. By setting $\epsilon_{\text{IP}} = \frac{1}{4\eta T}$, $\epsilon_{\text{norm}} = \frac{1}{2\eta T}$ and $\eta = \frac{1}{C^2\sqrt{T}}$, we obtain the desired bound.

The time complexity analysis is the same as that of Theorem 1.

