# OpenReview forum: "Quantum Algorithm for Sparse Online Learning with Truncated Gradient Descent"
_ICLR.cc/2025/Conference — Submitted to ICLR 2025_

### Official Review · Reviewer_has4 · 2024-11-01

**Soundness:** 3
**Presentation:** 3
**Contribution:** 2
**Rating:** 5
**Confidence:** 2

**Summary:**

In this paper, it presents quantum algorithms for studying the sparse online learning, and it can be applied to well-known machine learning problems, like logistic regression, the SVM, and least squares. This paper is built upon the seminal work of Langford et al. (2009), and it achieves a quadratic speedup in the time complexity: they get $\widetilde{O}(T^{5 / 2} \sqrt{d})$, where $T$ is the number of iterations and $d$ is the dimension of the data point. This time complexity improves from $O(Td)$ from Langford et al. (2009), under the assumption that $d > \widetilde{\Omega} (T^5)$ (for high-dimensional learning tasks). This time complexity is fairly compared with Langford et al. (2009) since they have the same regret $O(1 / \sqrt{T})$.

**Strengths:**

1. This paper gives a very clear presentation of the differences between their results and previous works. It applies the techniques of quantum amplitude estimation and amplification: ``quantum inner product estimation, quantum norm estimation, and quantum state preparation''. The improvement in time complexity is vital in our big data era, so the problem under the high-dimensional learning condition does match with the current focus.

2. Besides that, the three problems studied in this paper, logistic regression, support vector machine (SVM), and least squares, are very popular approaches used in machine learning. It is imperative to develop these as it has the potential to influence various fields.

**Weaknesses:**

1. However, it is unclear what the unique theoretical novelties of this paper are. While the authors provide a comprehensive overview of techniques adopted from other works, they lack a clear explanation of their own theoretical contributions. It is common for theoretical papers to defer proofs to the appendix. However, instead of merely stating the theorems, it is more important to present a high-level overview (or proof sketch) of how to establish these results, allowing readers to better understand the theoretical innovations of this work.

2. This paper misuses many instances of \citep and \citet.

3. While the size of datasets in practice continues to grow, and the authors cite numerous works (lines 74 to 77) that emphasize the significance of their contributions in the era of big data, it remains unclear under what circumstances, in practice, $d > \widetilde{\Omega}(T^5)$. I suggest that the authors include specific examples (with citations) where $d > \widetilde{\Omega}(T^5)$ is common in practice, as this would make the impact of their work more convincing.

**Questions:**

This paper uses the same assumption (Assumption 1) as Langford et al. (2009). Besides that, this paper also introduces Assumption 2. Is this assumption also used in Langford et al. (2009)? If not, then will this make the comparison being unfair?

---

### Official Review · Reviewer_cu1W · 2024-11-01

**Soundness:** 2
**Presentation:** 3
**Contribution:** 2
**Rating:** 5
**Confidence:** 2

**Summary:**

This paper develops new quantum online learning algorithms for logistic regression, SVM and least square problems. The work is based on the pioneering work of (Langford et al. 2009), utilizing a truncated gradient descent algorithm for sparse online learning. The developed method achieves the same regret bound while improving the time complexity from $O(Td)$ to $O(T^{5/2}\sqrt{d})$, allowing speedup when $d > O(T^5)$.

**Strengths:**

* The proposed method exhibits better time complexity when $d > O(T^5)$.
* Theoretical guarantees for the achievable regrets are provided.

**Weaknesses:**

* I'm not familiar with quantum algorithms, but to the best of my experience the data dimension $d$ and time step $T$ are usually at the same scale of magnitude for traditional deep learning, so I wonder how often will $d > O(T^5)$ actually happen in practice. I'll suggest the authors give some high-dimensional learning examples to motivate this improvement.

* There are no empirical results provided in the current version of the manuscript, which makes it hard for me to understand how much improvement could be brought by the method designed in this paper. I would consider raising my score if the authors could provide some empirical comparison between their method and the one in (Langford et al. 2009).

**Questions:**

Please see the weakness above.

---

### Official Review · Reviewer_gKgk · 2024-11-03

**Soundness:** 2
**Presentation:** 1
**Contribution:** 2
**Rating:** 3
**Confidence:** 2

**Summary:**

This paper developed a quantum sparse online learning algorithm by using truncated gradient descent. The time complexity is improved on the dependence on the data dimension. Applications to logistic regression, SVMs, and least squares are discussed.

**Strengths:**

The proposed algorithm achieves a reduced time complexity in the case the sample size $T$ is smaller than $\tilde O(d^{1/5})$, while attaining the same regret bound.

**Weaknesses:**

The presentation of the paper requires significant improvement. The flow lacks coherence, making it challenging to understand the paper's focus.

More importantly, it allocates substantial space to well-known models, such as regression and SVM, while providing limited attention to quantum computing and algorithms. The preliminaries on quantum computing are far from sufficient, which severely impacts the paper’s readability. For instance, $O_u|j\rangle|\bar{0}\rangle = |j\rangle |u_j\rangle$ in Section 3 and various notations/definitions in Section 4 are not accessible to readers in the machine learning field who might not have a background in quantum computing.

As someone who is not an expert in quantum computing, the paper’s lack of readability prevents me from confidently assessing the validity of the results or accurately evaluating its contributions. It is not clear to me whether the theoretical improvement is due to the quantum computing assumptions or the novelty of the proposed algorithm.

**Questions:**

- What does an exponential improvement in the dependence on $\\|u^\star\\|_2$ refer to on Page 4? The standard algorithm only has a polynomial dependence.
- Why does the gravity parameter $\alpha$ depend on the learning rate $\eta$?
- Does the speedup arise from the assumption that features and weights can be accessed in  $\tilde{O}(1)$ time coherently on a quantum computer, rather than the novelty of the proposed algorithm itself?
- Assumption 1 (ii) assumes $A,B \in \mathbb{R}_{>0}$ but $A$ or $B$ can be zero in the examples below.

---

### Official Review · Reviewer_g4kF · 2024-11-04

**Soundness:** 3
**Presentation:** 3
**Contribution:** 2
**Rating:** 5
**Confidence:** 2

**Summary:**

This manuscript presents a quantum sparse online learning algorithm for logistic regression, SVM, and least squares, building on the work of Langford et al. (2009). The authors demonstrate that the proposed algorithm achieves a quadratic speedup in time complexity, assuming efficient quantum access to inputs. While the manuscript provides substantial theoretical analysis, it lacks empirical evaluation to validate the conclusions, which makes the proposed method less persuasive.

**Strengths:**

The authors provide thorough theoretical analysis for the proposed method, which demonstrates greater practicality compared to the baseline approach. The research is foundational and has the potential to inspire other studies.

**Weaknesses:**

1. The manuscript lacks essential empirical studies, which reduces the persuasiveness of the proposed approach.

2. The baseline method is somewhat outdated, which could detract from the relevance of the proposed work.

3. The authors should strengthen the explanation of the research motivation to make the study more compelling.

4. The authors should provide an introductory explanation of quantum algorithms at the beginning of the article, and the related work section should cover recent advancements in truncated gradient descent and quantum algorithms.

5. There is discrepency in the notation for transposition and iteration (T), and a typo is present in Assumption 1.

**Questions:**

1.	Applicability to Nonlinear Models: Please clarify whether the method can be extended to nonlinear models. If not, specify its limitations.

2.	Input and Output of Algorithm 1: Please define the inputs and outputs of Algorithm 1 to facilitate understanding and reproducibility.

---

### Official Review · Reviewer_mUML · 2024-11-04

**Soundness:** 3
**Presentation:** 4
**Contribution:** 3
**Rating:** 6
**Confidence:** 1

**Summary:**

The authors study a quantum algorithm for online learning with a truncated version of gradient descent. This algorithm is already well studied in its classical form and going to its quantum counterpart one maintains the same regret rate, but with significantly smaller time complexity.

**Strengths:**

The article is easy to read and very clear even for someone not very knowledgeable about quantum computations.
I am not up to date in the literature, so I cannot comment on significance or originality.

**Weaknesses:**

The paper doesn't discuss how feasible it is to implement such an algorithm. Its advantage on the classical counterpart comes into play only for large $d$, which can make it harder to run this algorithm in practice. It also seem to me that running such an algorithm for large $T$ may accumulate a large number of errors.

**Questions:**

Can you give an intuitive explanation of the dependance between $T$ and $d$?

---

### Meta-Review · Area_Chair_BuVa · 2024-12-20

**Metareview:**

This paper presents a quantum algorithm for sparse online logistic regression. While reviewers found merits in the paper, the general agreement is that the novelty is limited by the fact that the authors method only improves on classical algorithms in a narrow parameter regime (very high dimensional data compared to number of samples). Since this is a purely theoretical paper, such an improvement might fit better at a more specialized venue on quantum algorithms or quantum ML.

**Additional Comments On Reviewer Discussion:**

The authors clarified minor technical misunderstandings and responded to many points about presentation issues. Most importantly, they engaged in a discussion about the parameter regime that this paper offers and improvement in, but ultimately did not succeed in successfully convincing me or the reviews that such a regime is of sufficient interest for a purely theoretical paper.

---

### Decision · Program_Chairs · 2025-01-22

Reject